# Phycobilisomes and Phycobiliproteins in the Pigment Apparatus of Oxygenic Photosynthetics: From Cyanobacteria to Tertiary Endosymbiosis

**DOI:** 10.3390/ijms24032290

**Published:** 2023-01-24

**Authors:** Igor N. Stadnichuk, Victor V. Kusnetsov

**Affiliations:** K.A. Timiryazev Institute of Plant Physiology RAS, 35 Botanicheskaya St., Moscow 127276, Russia

**Keywords:** chlorophyll, cryptophyta, cyanobacteria, endosymbiosis, glaucophyta, oxygenic photosynthetics, photosynthetic amoebae, phycobiliprotein(s), phycobilisome(s), red algae

## Abstract

Eukaryotic photosynthesis originated in the course of evolution as a result of the uptake of some unstored cyanobacterium and its transformation to chloroplasts by an ancestral heterotrophic eukaryotic cell. The pigment apparatus of Archaeplastida and other algal phyla that emerged later turned out to be arranged in the same way. Pigment-protein complexes of photosystem I (PS I) and photosystem II (PS II) are characterized by uniform structures, while the light-harvesting antennae have undergone a series of changes. The phycobilisome (PBS) antenna present in cyanobacteria was replaced by Chl *a/b*- or Chl *a/c*-containing pigment–protein complexes in most groups of photosynthetics. In the form of PBS or phycobiliprotein aggregates, it was inherited by members of Cyanophyta, Cryptophyta, red algae, and photosynthetic amoebae. Supramolecular organization and architectural modifications of phycobiliprotein antennae in various algal phyla in line with the endosymbiotic theory of chloroplast origin are the subject of this review.

## 1. Introduction

Cyanobacteria developed oxygenic photosynthesis 2.6–2.5 billion years ago (BYA) [1]. Approximately 1.9 BYA, after endosymbiotic uptake by a hetetrotrophic eukaryotic host of a cyanobacterium, possibly related to the present primitive genus *Gloeomargarita* and its evolution into a plastid, the first autotrophic eukaryote arose [2,3,4,5]. This one, common photosynthetic ancestor without a definite name (one possible name—Protoarchaeplastida), which has not survived to our time after primary endosymbiosis, underwent divergence into three eukaryotic lines [6,7]. According to the phylogenomic data, first, 1.7–1.6 BYA, Glaucophyta (blue branch) was separated from the common evolutionary tree [8]; later, 100–200 MYA (million years ago), red algae, or Rhodophyta, (red branch) emerged from the remaining common evolutionary stem of red and green algae; and finally, 1.1 BYA green algae appeared [9]. These three groups of photosynthetics, collectively called Archaeplastida (Figure 1A) and referred to as Plantae have double membrane plastids and coexist in the biosphere, together with cyanobacteria, up to the present time. In the subsequent events of secondary endosymbioses, green and red microalgae gave rise to photosynthetics with four-membrane plastids (reduced to three membranes in some algae), resulting in various new algal phyla [10]. Together with the three Archaeplastida groups, there are approximately ten major algal taxa. Higher plants inherited the Archaeplastida chloroplasts from the green algae [9].

The monophyletic origin of eukaryotic photosynthetics has one exception. The plastids of a small group of amoebae that explote photosynthesis originated later than Archaeplastida from an ancestral cyanobacterium of the genus *Synechococcus* (Figure 1B). Apart from this fact, in the cells of some dinophytic algae, their own secondary plastids were subsequently lost and replaced by the plastids of the engulfed cryptophytic or ciliate algae. This effect has been called tertiary endosymbiosis [6,9,10]. During the repeated emergence of new groups of algae, the two photosystems, PS I and PS II, as the basis of oxygenic photosynthesis, started from cyanobacteria and remained unchanged, but the light harvesting cyanobacterial antenna represented by phycobilisomes (PBSs) underwent changes up to replacement by other antennas, Chl *a/b* or Chl *a/c*, as pigment–protein complexes [11,12].

Due to the extraordinary features of light absorption in a wide spectral range and the labile molecular architecture, PBSs or individual phycobilproteins have been preserved in red, glaucophytic, and cryptophytic algae as well as in photosynthetic amoebae. The goal of this communication is to consider the features of PBS antennae in the listed groups of photosynthetics. The purpose of this review was to summarize and consider all known cases of phycobiliprotein antennae in relation to the evolutionary origin of different groups of oxygenic photosynthetics. We also outline possible reasons why phycobiliproteins are absent in terrestrial plants and many groups of algae.

## 2. Phycobiliproteins and Phycobilisomes of Cyanobacteria

There are approximately 1500 known species of unicellular, filamentous, and colonial cyanobacteria which are the only prokaryotes that perform oxygenic photosynthesis. Brilliantly colored phycobiliproteins serve as antennae in the pigment apparatus of cyanobacteria, supplying absorbed light energy to PS I and PS II. These chromophorylated water-soluble proteins consist of 16–18 kDa *α* and 17–20 kDa *β* polypeptides in a 1-to-1 ratio. Such (*αβ*)_1_-heteromonomers spontaneously aggregate into (*αβ*)_3_-trimers and then into (*αβ*)_6_-hexamers [13]. Four different open-chain phycobilin chromophores, namely, phycocyano-, phycoerythro-, phycourobilin, and cryptoviolin (phycoviolobilin), covalently bind in different ratios via a thioether bond to the Cys residues in both polypeptide subunits. Each *α*-subunit attaches one or two chromophore groups; each *β*-polypeptide binds one to four tetrapyrrole chromophores, depending on the specific phycobiliprotein. Each phycobiliprotein has chromophores of one (PC, APC, C-PE) or two different varieties (R-PC, PEC, CU-PEs, and R-PE). A unique trichromatic-variety phycocyanin, R-PC V, was revealed in oceanic *Synechococcus* sp. WH8102 [14]. As a result, approximately 15 different phycobiliproteins have been described (Table 1). 

Colorless linker proteins of about 30 kDa bind disc-shaped hexamers and trimers into cylindrical rods, and these, in turn, assemble into PBSs consisting of two main parts: the APC core and several peripheral rods of other phycobiliproteins surrounding the central core. PBSs of the vast majority of cyanobacteria are hemidiscoidal. Three morphological types have been observed: (1) with a two-cylindrical core and six fan-shaped lateral cylinders; (2) with a three-cylindrical core and six cylinders; (3) with a five-cylindrical core (three major plus two semicylinders) and eight lateral cylinders (Figure 2A–C).

Bundle-shaped PBSs of the genus *Gloeobacter* have a somewhat unusual architecture with only vertical lateral cylinders (Figure 2D). This morphology saves space on the membrane surface, since cyanobacteria of this genus have no thylakoids and all PBSs are located on the cytoplasmic membrane [15]. When, for some reason, altered conventional PBSs in mutant cyanobacterial cells cannot properly attach to the thylakoid membrane, the energy feeding of PS I is provided by newly synthesized small cylindrical PBSs anchored to PS I by the special CpcG2 polypeptide linker [16,17]. Some cyanobacteria, such as *Calothrix* sp., can vary their PC and/or PE ratio in PBS depending on the light quality. This phenomenon, realized in several variants, is known as complementary chromatic adaptation (CCA) [18].

In the PBS, absorbed solar energy is funneled from the lateral cylinders to the core and eventually to the terminal longwavelength emitters—chromophorylated ApcD and ApcF varieties of APC subunits and the ApcE (Lcm) and chromophorylated large core-membrane linker proteins located in the bottom of APC cylinders. From collectively acted terminal emitters, the energy rapidly rolls to PS I and PS II [19,20,21]. Regarding PS II, there has long been a general consensus that in the plane of the thylakoid membrane, PBSs are disposed on the outer surface of PS II dimers [22]. 

The situation with PS I is more complicated [23,24,25,26]. In cyanobacteria, PS I mainly forms trimers with a certain share of monomers [27,28]. In contrast to the PS II dimers [22], the surface of PS I shows a major protrusion of three hydrophilic polypeptide subunits (PsaC, PsaD, and PsaE), which extends into the cytoplasm [27,28] and prevents a tight binding of PBS to PS I trimer due to its threefold symmetry. Computer modeling demonstrated the docking of PBS to the PS I monomer only, corresponding to a realization of a highly likely energy transfer between these two pigment–protein complexes [29].

Along with the “common” PBSs and the “common” cyanobacteria, there are other species that have caught everyone’s attention.

### 2.1. Prochlorophyta: Chlorophyll b-Containing Cyanobacteria

Prochlorophyta, or Prochlorales, is a mixed group of cyanobacteria containing Chl *a/b*-proteins instead of PBS as a photosynthetic antenna. Few known Prochlorophyta members form three genera, namely, *Prochlorococcus, Prochloron*, *and Prochlorothrix*, with cells having green or greenish color. The genus *Prochlorothrix* is represented by two filamentous freshwater species, *P. hollandica* [30] and *P. scandica* [31]. *P. didemni*, the only one known species of the *Prochloron* genus, is an obligate symbiont of the didemnid ascidia *Lissoclinum patella* [32]. Various described ecotypes of the genus *Prochlorococcus marinus* inhabit different geographical areas of the ocean and belong to the most abundant representatives of oceanic photosynthetic picoplankton.

Light-harvesting membrane Chl *a/b*-binding proteins in Prochlorophyta are derived from the CP43 polypeptide of PS II and are named prochlorophyte Chl-binding (Pcb) proteins [33] or, less frequently, designated CBP [34], while the Chl *a/b*-containing antenna in the pigment apparatus of Viridiplantae consists of LHC-polypeptides. According to this type of antenna, Prochlorophyta have stacked regions of thylakoids (Figure 3), making them different from other cyanobacteria [35]. Pcb-proteins form in thylakoid membrane supercomplexes with both photosystems. The PS I trimers are surrounded by a single ring of Pcbs, while dimers of PS II are in contact with two or three divided by a gap Pcb arches [36]. Interestingly, cells of *Prochlorococcus* contain their divinyl derivatives with the same absorption properties instead of Chls *a* and *b* [37]. In addition to the presence of Chls *a* and *b*, very small amounts of Chl *c* variety in Pcb of *P. didemnii* also cannot be excluded [38].

Of all Prochlorophyta, *Prochlorococcus* attracts the most attention because of its prominent role in marine primary productivity and the presence of PE [39,40], in addition to its Pcb-antenna proteins. Cyanophages infecting *Prochlorococcus* contain genes for phycobilin-synthesizing enzymes, and these are expressed in *Prochlorococcus*, raising further questions of how the PE genes could have been acquired by this species [41]. Various described marine strains of *Prochlorococcus* belong to one of two ecotypes that are specifically adapted to either low-light or high-light conditions [42,43]. Low-light-adapted ecotypes carry a gene cluster with the *cpeA* and *cpeB* genes of PE to produce a small amount of a functional light-absorption pigment. Differences in the sequences of the *α* and *β* polypeptide chains and the revealed presence of only one phycourobilin-like attachment site on the *α* subunit distinguish this red pigment from the known PEs of PBS-containing cyanobacteria. However, the supramolecular organization and aggregation state of PE molecules due to their very low content in thylakoids remain unknown. According to electron micrographs of immunogold-labeled cells, PE is attached to the thylakoid membrane at its lumen side [44].

In the genome of the high-light-adapted strains, only a single and free-standing *cpeB* gene occurs. This gene encodes a derived form of *β*-PE, the function of which has remained enigmatic thus far [45]. Interestingly, degenerated forms of the *α* and *β* subunits of other cyanobacterial PEs without perturbing their folding, structural stability, and fluorescence functionality were demonstrated to be stable in vitro [46]. Correspondingly, phycobilins are known to serve as single chromophores in cyanobacteriochromes [47]. Therefore, such a degenerated form of PE could change its purpose to being a photoreceptor pigment in *Prochlorococcus.*

### 2.2. Chlorophyll d-Containing Cyanobacteria, Acaryochloris marina

*Acaryochloris marina* is a widespread free-living cyanobacterium that exploits special marine habitats depleted in visible solar radiation and characterized accordingly by an increased proportion of infrared penetrated light. These shady sieve site conditions fully correspond to the microlayer of seawater beneath the body surface of didemnid ascidians, where this species was first described in 1996 [48]. Since then, numerous free-living *Acaryochloris* strains have been reported to be distributed in various environments, including marine stromatolites, saline lake epilithic biofilms, and some special freshwaters [49]. 

The pigment apparatus of *A. marina* is characterized by the presence of Chl *d*, with absorption shifted to the long-wavelength spectral region compared to Chl *a*. Different kinds of Chls can readily be distinguished by their S_0_ → S_1_ long-wavelength absorption peak positions. This feature helps gain insight into the evolved mechanism for the realization of infrared region photosynthesis. The polypeptide compositions and aggregation states of both the PS I and PS II core complexes (dimers of PS II and monomers/trimers of PS I) of *A. marina* are similar to those of well studied Chl *a*-containing cyanobacteria, but in contrast, each photosystem uses Chl *d* as the main photosynthetic pigment, including the special chlorophyll pairs in both their RCs [50]. The light-harvesting antenna of *A. marina* is a special Pcb-protein of ca. 34 kDa incorporated by Chl *d* in concert with some part of Chl *a*, indicating that this membrane antenna is similar to the Chl *a/b*-protein of prochlorophytes [51]. Pcb-proteins encircle in the form of a ring the PS II dimers, but data on Pcb composition and function in PS I are currently absent. In summary, Chl *d* can account for ~95% of total Chls (*a* + *d*) in the contents of PS I and PS II [52]. 

All the known strains of *Acaryochloris* are free from phycobiliproteins except the first described and most extensively studied “under ascidian niche found” *A. marina* MBIC11017. Classic cyanobacterial PBSs in this strain are eliminated. Instead, *A. marina* MBIC11017 contains the rod-shaped phycobiliprotein aggregates located on the cytoplasmic side of the thylakoid membranes along with the intrinsic membrane Pcb-antenna. Each rod is composed of four phycobiliprotein hexamers equal in size to lateral rods in a typical PBS from other cyanobacteria. Rods consist of PC (*αβ*)_6_-hexamers with some admixture of pigment, whose absorption spectrum resembles that of APC. The complete chromosome sequence of *A. marina* demonstrates that, along with PC-related genes, only the *apcB* gene is present for coding the *β*-chain of APC, while the corresponding *apcA* is absent [52]. This means that very unusual (*αβ*)_1_-heteromonomers mixed from *α*-PC and *β*-APC polypeptides most likely compose the last trimer in the bottom hexamer of a cylinder adjoining the outer thylakoid membrane surface. The presence of both PC and APC, however, within one hexamer seems to be a unique feature of the phycobiliprotein aggregates in *A. marina*.

The architecture of native cylindrical phycobiliprotein aggregates is much less complex than that of complete PBS. However, the energy transfer to Chl *d* is very efficient and takes about 70 ps [53]. In isolated preparations [54,55] and in photosynthetic action spectra [56], it has been found that the biliproteins are physically attached to the PS II complexes. Comparison of the surface areas for the rounded bottom of phycobiliprotein cylinders and PS II shows that each PSII dimer can anchor one or two cylinders. Currently, it is not clear whether such an attachment needs special linker protein(s) or not. Due to the cytoplasmic surface area of the PS II dimer [22], there is enough place for the ends of two cylinders. Dimers of PS II in *A. marina* are paired in tetramers [50]. As a consequence, the phycobiliproteins form separate patches on the thylakoid surface that, highly probably, consist of four cylinders feeding by absorbed energy these PS II tetramers. 

Complete sequencing of the *Acariochloris* genome showed that phycobiliprotein genes are not located in the chromosome but in large cell plasmids. This led to the conclusion that genes for PBS appear to have been previously lost in the common ancestor of *Acaryochloris* and some other marine cyanobacteria. *A. marina* MBIC11017 is the only *Acaryochloris* strain which is thought to have reacquired a set of the phycobiliprotein genes via HGT [57].

### 2.3. Chlorophyll f-Containing Cyanobacteria and Their Involvement in the Process of Photoacclimation to Far Red Light

Chl *f* was discovered as a minor pigment in a filamentous cyanobacterium *Halomicronema hongdechloris* collected from living columnar marine stromatolites [34] and has since been found in a number of other cyanobacteria, both filamentous and uni-cellular, including terrestrial species [58]. Compared to *A. marina*, Chl *f*-containing cyanobacteria can photosynthesize further into the far-red and near-infrared regions (Chl *f* gains enhanced absorbance by up to 800 nm) [59]. The amount of Chl *f* in white light is very small or absent [60], and its biosynthesis is induced only under FaRLiP (far-red light photoacclimation) conditions. In contrast to Chl *d* in *A. marina*, studies performed over the past 10 years have shown that Chl *f* is not part of a special pair in both RCs and functions only in a light-harvesting role in the core antennae of PS I and PS II. The precise cryo-EM data have shown that the FaRLip-PS II pigment–protein complex from *Synechococcus* sp. PCC 7335 binds one Chl *d* and four Chl *f* in place of several of 35 Chl *a* molecules present in PS II when grown in visible light [61]. Analogously, the high-resolution structure of the far-red PS I core from *H. hongdechloris* showed that in far-red light, it binds 83 Chl *a* plus 7 Chl *f* sassociated at the periphery of the PS I core antenna [62]. Simultaneously with these pigment replacements, several polypeptide subunits of PS I and PS II cores are remodeled using subunits paralogous to those produced in white light [61,62]. In doing so, even then Chl *f* constitutes only a small part (less than 10%) of the remaining predominant Chl *a* but expands the spectral region of solar light absorption. 

The architecture of PBS from *Leptolyngbia* sp. strain JSC-1 exhibits an extensive response to growth in far-red light simultaneously with the stimulation of Chl *f* biosynthesis. The hemidiscoidal pentacylindrical PBS of this species corresponds to the one of varities of chromatic adaptation synthesizing PE in green and white light and the PC only under red light conditions. In far-red light, the PBS core diminishes to only two APC cylinders, retains PE disks in lateral cylinders while absent in red light, and produces another ApcE linker (ApcE2 instead of ApcE1) with noncovalently bound terminal phycobilin chromophor [59]. This longwavelength-shifted terminal chromophore better corresponds to effective energy transfer from PBS to Chl *f*-modified complexes of PS I and PS II. It has been suggested that *apcE2* could be used as a marker of FARLiP adaptability in different tested species of cyanobacteria [63]. The genome of *Leptolyngbya* sp. JCC-1 has a 21-gene cluster that encodes paralogues of most of the core polypeptides of PS I, PS II, and PBSs; similar clusters are found in other FaRLiP-cyanobacteria [59]. All photoacclimation processes usually take approximately one day. The restructured pigment apparatus (i.e., PS I, PS II, PBS) acquires the ability to absorb at longer light wavelengths, which enables cyanobacteria to grow photoautotrophically in far-red light. 

To summarize somewhat, it is stated that changes in the composition of phycobiliproteins occur differently in cyanobacteria possessing various kinds of Chl pigments. There is a complete loss of PBSs and phycobiliproteins in Chl *b*-containing phototrophs, a reduction from whole PBS to phycobiliprotein cylinders in Chl *d*-containing species, and only a partial rearrangement of the PBS core and replacement of individual phycobiliproteins in Chl *f*-containing cyanobacterial cells. 

## 3. Red Algae Phycobilisomes

Red algae (Rhodophyta) are represented by six thousand species, the vast majority of which are marine macrophytes. The number of unicellular algae, including freshwater inhabitants, is small—approximately 150 representatives. Red algae are divided into two branches: Cyanidiophytina and Rhodophytina (Figure 1A). The first consists of only a dozen microalgae that live in hot sulfur springs and are the only known group of polyextremophiles among all eukaryotic organisms. The thermoacidophilic Cyanidiophytina with a single class of Cyanidiales were separated 1.4 BYA from the rest of the red algae. Together with Glaulaucophyta, they are the most ancient eukaryotic photosynthetics [11,64].

Cyanidiales inherited from cyanobacteria one of the simplest varieties of PBS—a semidiscoidal PBS with a tricylindrical core and only two types of pigments: blue PC of the lateral cylinders and light-blue APC in the PBS core. Therefore, the organisms called red algae demonstrate an unusual blue-green cyanobacterial coloration in this case. PBSs of Cyanidiales, which have many overlaps in their protein composition, chemical structure of chromophores, and general morphology with cyanobacteria (Figure 3), provide convincing evidence for the theory of chloroplast endosymbiosis. Unicellular seaweeds, such as *Rhodella violacea*, also contain hemidisoidal PBSs, but they already include the red PE pigment [65].

A feature of red algae is the appearance of B-PE and R-PE. Both pigments have a total of five phycobilin PEB and PUB chromophores on (*αβ*)_1_ monomers plus chromophorylated *γ*-subunits derived from linker polypeptides. The B-PE variety without the *γ*-subunit is denoted as b-PE, whereas R-PE always contains the *γ*-subunit. Depending on the number of chromophore groups on the *γ*-subunits and various combinations of the PEB and PUB chromophores, there are at least three kinds of this pigment [66].

Macrophytic red algae contain PBSs of maximum size. For their external formes, they were called hemi-ellipsoidal [67] and block-shaped [68]. Because of the apparently chaotic mutual arrangement of the lateral cylinders and the lack of strict geometry, they were inaccessible for structural studies until recently. Only in the most recent years has significant progress been made here. Using single-particle cryo-electron microscopy, details of the structure of hemilipsoidal PBS of the seaweed *Porphyridium purpureum* [69] and block-shaped PBS of *Griffitshia pacifica* [70] were investigated. The architectural features of these PBSs, due to their great complexity, can be visualized only when shown from different angles in several drawings, but due to the importance of the obtained results, they should be listed, noting the most important details.

The general structure of PBS, which has two parts, the core and lateral cylinders, remains the same as in cyanobacteria, but becomes as complex as possible. In *P. purpureum* the molecular mass of PBS is equal to 14.7 MDa. One PBS contains 1600 (!) phycobilin chromophores. The number of lateral cylinders increases to 14. They contain 528 PE and 72 PC, not counting linker proteins. The length of the cylinders varies, from a single PE hexamer to a series of docked hexamers. The size of the PBS core is reduced to 8 trimers of APC. Block-shaped PBSs of the *G. pacifica* turn were even larger. The mass of PBS reaches 18.0 MDa, and the total number of chromophores increases to 2048. The PBS consists of 862 polypeptides. Some of the lateral cylinders consist only of R-PE hexamers. Therefore, PE makes up to 90% of the phycobiliproteins in the PBS, providing colouration of macrophytes.

Numerous studies primarily performed with Cyanidiales have shown that PS II in red algae, such as in cyanobacteria, forms dimers [71], but PS I, unlike cyanobacteria, exists in thylakoid membranes only as monomers [72]. The formation of PS I trimers is prevented by an additional membrane antenna that has appeared in the red algae in the form of Chl *a*-containing LHC-proteins, related to the Chl *a/b*-containing antenna of Viridiplantae. Information on the membrane status of PS I and PS II is very important for upcoming studies on how giant PBSs realize energy transfer to the pigment–protein complexes of the photosystems. 

The necessity of the cell’s expenditure for the biosynthesis of giant phycobiliproteins also correlates with the effect of reducing gene content.

## 4. Glaucophytic algae

Glaucophyta or Glaucocystophyta is a division of biflagellate unicellular algae consisting of one class of Glaucocystophyceae, several genera, and according to a combination of data from various sources, 26 described species. Flagella in colonial forms are reduced. The search for marine creatures is not ruled out, but all described species are apparently limited to freshwater environments, including small marsh populations. Cells have photosynthetic organelles, formerly called cyanelles and even ”living fossils” but later renamed muroplasts [73], since their origin as plastids is no longer in doubt. Muroplasts maintain several biochemical and structural features of cyanobacteria, among which the peptidoglycan layer and carboxysomes, organelle-like polyhedral bodies involved in CO_2_ fixation, are of particular importance [74,75]. These relic plastid substructures provide compelling evidence for the endosymbiotic theory being later independently lost in rhodoplasts and in the Viridiplantae branch of Archaeplastida. Cyanobacteria posses a 30 nm mesh of peptidoglycan (monosaccharides cross-linked by peptide chains) surrounding the plasma membrane that is involved in cell preservation and cell division. The retained vestiges of cyanobacterial peptidoglucan are located between the two membranes of glaucophyte plastid and play an important role at the start of its division (Figure 3). Carboxysomes, organelle-like polyhedral bodies, are involved in cyanobacteria in CO_2_ fixation. Similar to red algae, glaucophytes form storage glucan in the cytosol [76]. 

*Cyanophora paradoxa* is a single-celled nitrogen-fixing Glaucophyta member with a sequenced genome that has been used for many years as a model organism in various laboratories. Accumulated data have regularly reflected in a number of reviews [73,74,75,77,78,79]. *C. paradoxa* has one or two plastids with nonstacked thylakoids, as also occurs in red algae. PS II particles have been isolated from *C. paradoxa* thylakoids and characterized with respect to their oxygen-evolving activity, spectral characteristics, and polypeptide content [80,81,82]. In particular, the 12 kDa protein PsbU of PS II present in cyanobacteria and red algae was not detected [82]. Because many muroplast genes have been displaced by endosymbiotic gene transfer (EGT) to the nuclear genome [83], biochemical studies and comparative genomic data may not be sufficient to identify all the components and molecular structures of the PS II complex(es). Nevertheless, despite the current lack of cryo-electron microscopic data, there is no reason to doubt that PS II in glaucophyte thylakoids is organized as dimers, similar to other known photosynthetics. Glaucophyta do not contain LHC-type antennae [83], which proteins prevent by attachment to the outer surface of PS I monomers and their further aggregation in plastid membranes of the red algae. In accordance with this, it was shown that PS I in *C. paradoxa* consists of not only monomers [84] but also tetramers in thylakoid membranes [85]. Tetramers are highly specific and loosely organized as dimers of dimers, unlike tetramers found in some species of cyanobacteria, such as *Anabaena* [86].

The blue-green coloration of *C. paradoxa* and other known glaucophytes is due to the presence of only APC and PC in their PBSs and coincides with the coloration of the red microalgae Cyanidiales and many cyanobacteria whose cells do not contain PE. PBSs of *C. paradoxa* can be distinguished as rows on the cytoplasmic surface of thylakoid membranes with the assumption that PBSs are docked to PS II dimers [87]. The chromophorylated polypeptides of APC and PC and anchor Apc E proteins of PBS are encoded in *C. paradoxa* plastom, but uncolored linker genes due to EGT are located in the nuclear genome [74,88]. The typical rod-linker protein CpcC, involved in the formation of PC rods of cyanobacterial and red algal PBSs, has not been found in *C. paradoxa*. This suggests that the novel *Cyanophora* proteins CpcK1 and CpcK2, which harbor PC linker domains, play the role of bundling the PC hexameric disks in lateral cylinders. Phylogenetic analyses of these two rod linkers indicate that these proteins are products of gene duplications, which probably occurred only in glaucophytes [88]. Currently, the pigment apparatus needs additional investigation. 

## 5. Photosynthetic Amoeba *Paulinella*

*Paulinella* is a *genus* of at least eleven ameboid species with both marine and freshwater representatives (*Clade* SAR) [89]. There are three photosynthetic representatives in this group of unicellular organisms, with the most studied being filose thecamoeba *Paulinella chromatophora*. The blue-green colored plastids of *Paulinella* are usually named chromatophores or cyanelles, as in glaucophytic algae. Chromatophores of evolutionary cyanobacterial origin provide photoautotrophy, unlike the closest *Paulinella* ameboid heterotrophic relatives that simply feed on cyanobacteria via phagocytosis [90]. The *Paulinella* chromatophore is monophyletic with α-cyanobacteria (the *Prochlorococcus/Synechococcus* cyanobacterial clade), which was engulfed by a heterotrophic ameboid and diverged to photoautotrophy from its sister species 90–140 ~MYA [91,92,93]. It is the only known case of primary endosymbiosis that evolved from a cyanobacterium other than that determined for Archaeplastida, a β-cyanobacterial ancestor.

Although the *Paulinella* chromatophore retains key cyanobacterial features such as a bacterial peptidoglycan cell wall, carboxysomes (cellular compartments that house ribulose-1,5-bisphosphatecarboxylase/oxygenase), and PBSs (all the listed properties are the same as in Glaucophyta), it also has morphological traits that are consistent with a photosynthetic organelle status such as its inability to survive outside of the host cell, its residence in the host cytoplasm, not being encapsulated within a vacuole, coordination of chromatophore number, and division with host cell division (i.e., two chromatophores per mature host cell). More convincing are the findings that the chromatophore genome is greatly reduced in size (~1 Mbp) and gene content (ca. 850 protein-coding genes) relative to free-living α-cyanobacteria (generally 2–5 Mbp, ca. ~3000 protein-coding genes) [94].

Because *Paulinella* acquired oxygenic photosynthesis independently of Archaeplastida, it represents the most valuable source for understanding early endosymbiothic events that are poorly understood [94,95]. Al genes essential for the function of PS I and PS II, the cytochrome b_6_/f complex, the F-type ATPase, and photosynthetic electron transport are present on the chromatophore genome. A complete set of genes for APC and PC as well as linker proteins of the PBS are also present [92]. On the slit sections of *Paulinella* chromatophores, interthylakoidal particles of 30 to 40 nm diameter are observed which, by analogy with the photo of cyanobacterial membranes, can be considered an image of PBSs [90]. Unfortunately, *Paulinella* is very difficult to grow in the laboratory, and the presence of cell-coating theca makes it difficult to use preparative biochemistry. Most of the obtained knowledge is based on genetic studies [96]. It is possible only to state that there is a pigment apparatus with PBS, the two photosystems, and typical photosynthetic reactions [97]. Concrete biochemical and biophysical data on photosynthesis to compare them with eco-morphological information and the results of genetic studies will hopefully follow in the future.

## 6. Cryptophyte algae

Cryptophytes, Cryptophyte algae, or Cryptomonads have biflagellate asymmetrical cells with a size not exceeding 50 μm. They inhabit fresh and marine waters and include approximately 20 genera and more than 150 species with variable morphology. Many cryptophytes are photosynthetic; mixotrophic and colorless heterotrophic species with reduced plastids are also known. These fragile microalgae, which easily lose their flagella, are the subject of regular reviews [98,99,100,101,102,103,104,105].

*Cryptophytes* belong to the diverse group of Chromoalveolata whose chloroplasts were inherited from an engulfed red algal cell by an unknown eukaryote and have been surrounded by four membranes (Figure 4). In cryptophytes, the reduced nucleus, or nucleomoph, with three chromosomes, which remains from secondary endosymbiosis with a red alga, is present between the two pairs of envelope membranes. This means that, including the DNA of the plastome and mitochondria, the cells contain four separate genomes.

Chl *a/c* containing light-harvesting complexes of Cryptophytes and other clades of Chromoalveolata are part of the family of CCA-proteins (Chl *a/c*-containing proteins) [106,107]. Cryptophytic algae differ from all other known chromoalveolates due to their unique complement of phycobiliproteins to Chl *a/c*-proteins in antennae functions. The phycobiliproteins of cryptophytes do not repeat those known for cyanobacteria and red algae, being only inherent pigments. By their coloration, pigments are summarized in red PEs and blue PCs. In any one algal species, there is one of eight (possibly nine [102]) unique for cryptophyte algae phycobiliproteins that are designated by their longwavelength absorption peak positions (in nm) as: Cr-PE545, Cr-PE555, Cr-PE566, Cr-PC569, Cr-PC577, Cr-PC612, Cr-PC630, and Cr-PC645 [98,99,102]. The color of these pigments and the shape of the absorption spectra are determined by different combinations of chromophores and can also arise from sequence differences in the polypeptide subunits [108] of each phycobiliprotein. Cr-PE545, according to the evolutionary history of Cryptophyta, is an ancestor for this branching line of piments [109]. 

Of the six chemically identified sorts of phycobilin chromophores found in these phycobiliproteins, phycoerythrobilin and phycocyanobilin coincide with those known for red algae and cyanobacteria; the remaining four are unique to cryptophytes [99]. Each phycobiliprotein exists in the form of polypeptide heterodimers (*α*_1_*βα*_2_*β*), i.e., it contains two different *α*-polypeptide subunits and two identical *β*-polypeptides with a relatively small mass of ∼60 kDa [99,102,110,111]. The *α* subunit evolved from a recently identified family of red algal PBS scaffolding proteins that stabilize the *β* polypeptides of PE in lateral cylinders of red algal PBSs [112]. The *β* subunit descended directly from a red algal PE *β* subunit [113]. 

The *α* polypeptides have one chromophore each, the *β* polypeptide includes three phycobilin chromophores, and in total, the dimer contains eight chromophore groups. The *β* subunits are encoded in the plastid genome, but *α*_1_ and *α*_2_ are encoded by the cell nucleus and are transported from the cytoplasm through four outer membranes and the inner thylakoid membrane inside the chloroplast, where they assemble into dimers. With only two genes for the *β* subunit [114], repeated duplication resulted in a large family of *α* subunits including up to 20 genes, although any two of them are preferentially expressed at each moment depending on the light conditions [115]. The biosynthesis of subunits is under strong genetic control. Insertion of a single aspartic acid residue of *α* subunits in Cr-PC612 of *Hemiselmis virescens* results in distinct quaternary conformations of the (*α*_1_*βα*_2_*β*)-dimer [116]. In contrast to PBSs, the heterodimers are not localized on the stromal surface of the thylakoid membrane but occupy the entire space of the lumen and are most likely assembled into cylindrical structures oriented perpendicularly to the thylakoid membrane [110,117]. To date, all attempts to obtain in vitro phycobiliprotein macrostructures larger than the heterodimers to resolve their assembly have failed [111]. 

The isolation of thylakoid membrane fragments followed by cryo-EM microscopy demonstrated that antennal Chl *a/c*-proteins are connected in thylakoids with the PS I monomers and PS II dimers [118,119]. Studies of energy transfer pathways from the phycobiliproteins to PS I and PS II by steady-state and time-resolved fluorescence measurements and some other techniques made in several years with different algae demonstrated that phycobiliproteins are primarily responsible for funneling energy to PS II [120,121,122]. The absence of an energy transfer pathway from the phycobiliproteins to PS I in cryptophytes would not correlate with the corresponding data obtained for PBSs of cyanobacteria and red algae. In some other studies it was postulated that phycobiliproteins sensitize both PS II and PS I [123,124]. Several factors did not allow us to draw the definite conclusions about the possible association between phycobiliproteins and photosystems. In our opinion, the data obtained using the photosynthetic action spectra of the two photosystems are the most convincing [125,126]. An advantage of this method over the others is that the shape of the action spectrum containing all pigment bands of each photosystem repeats the spectrum of its absorption in vivo and is not influenced by the presence of the other photosystem [126]. Cr-PE545 from *Rhodomonas lens* is responsible for funneling energy to PS II only [125,126]. These differences in the presented data should not be confusing since it is known that in cryptophyte cells, the phycobiliprotein content strongly depends on the light conditions of their growth and other factors. Corresponding estimates of the direction and rate of energy transfer may vary. Obviously, additional experiments and measurements are needed here.

In addition to ambiguities with phycobiliprotein involvement in energy transfer, the presence of those or other Chls *c* in cryptophyte algae also requires clarification. Several pigments of the Chl *c* family are known: Chl *c*_1_, Chl *c*_2_, Chl *c*_3_, etc. Chl *a/c*-proteins in Cryptophyta are believed to contain Chl *c*_2_. This view relies largely on Jeffrey’s original work, performed in 1976, where Chl *c*_2_ was found in nine different species of cryptophyte algae [127]. Apparently, Chl *c*_2_ is indeed the predominant pigment in various Cryptophyta species. However, when the Chl *c* composition was analyzed in two cryptophyte algae, *Rhodomonas maculata* Butcher and *Chroomonas* sp. Hansgirg, Chl *c*_2_ was detected only in the former, while the latter contained Chl *c*_1_. In addition, the other pigment of this group, MgDVP (Mg-2,4-divynylpheoporphyrin a_5_ monomethylester), was identified in both algae in minor amounts [128]. Obviously, the situation with Chl *c* is not so simple, and the composition of Chl *c* should be determined separately for each species. Of particular interest is the presence of Mg-DVP as a part of a Chl (*a/b*/Mg-DVP) light-harvesting protein in *Prochloron* sp. [129].

One can conclude that the phycobiliprotein part of the cryptophyte pigment apparatus differs greatly from cyanobacteria and other compared algae in the absence of PBS, the nonformation of aggregates larger than dimers in solution, the anchoring to the lumenal side of the inner thylakoid membrane, the large number of different phycobilins, the diversity of phycobiliproteins themselves, and, highly likely, the absence of interaction with PS I. All these features have yet to be elucidated. 

## 7. Tertiary Endosymbioses Involving *Cryptophyte algae*

Endosymbiosis is a very efficient evolutionary strategy that greatly increases the diversity of microalgae. This phenomenon is not limited to primary and secondary endosymbioses, but extends to tertiary endosymbiosis, primary serial symbiosis, and kleptoplastia demonstrating distinct levels of host–endosymbiont integration (see [5,6,10,130,131]. The case of tertiary endosymbiosis seems to be the most significant in terms of the changed pigment apparatus and its direct involvement in photosynthesis. This is the event when a unicellular alga lacked inherited plastids after secondary endosymbiosis and reacquired them from the ingested microalgae of other groups.

Tertiary endosymbiosis was revealed for a number of dinoflagellate species [132]. Dinoflagellates (alternative names: Dinophyta, Pyridenea, Pyrrophyta) include approximately 2500 modern members belonging to the superclade Alveolata. These shell-covered flagellates have fantastic trophic capabilities. Approximately half of all species are photoautotrophic using secondary plastids of red algal origin. There are also colorless heterotrophs that lack plastids. Many predatory dinoflagellates practice kleptoplastia—the temporary preservation of “stolen” plastids of microalgae chloroplasts, which they feed on. Mixotrophy, i.e., a combination of phototrophic and heterotrophic metabolism, is also widespread [133]. 

Dinoflagellates have only three membranes surrounding their plastids due to the reduction of the outer fourth membrane, which is considered a progressive feature facilitating the transmembrane transport of proteins and assimilates. Chromoplast thylakoids in stacks of three, containing the Chl *a/c*-protein and peridinin as the main pigment, are regarded as the original dinophyte plastids [6]. The propensity for plastid loss and replacement was demonstrated for various dinoflagellate species. Diatom-like, haptophyte-like, prasinophyte-like, and cryptophyte-like plastids were demonstrated in dinophyte species (summarized in [134]). 

We focus on tertiary plastid endosymbiosis with a cryptophytes origin found in *Dinophysis* genus (Figure 4). Phycobiliproteins are not a native group of antennae for Dinoflagellates, but it has been demonstrated that the absorption and fluorescence spectral characteristics of three *Dinophysis* species indicate their presence [135]. The presence of phycobiliproteins has been confirmed by immunogold phycoerythrin localization in chloroplasts [136]. The chloroplasts greatly resemble cryptophycean chloroplasts having paired thylakoids and electron-dense material inside the thylakoid lumen. They are bound by only two membranes [137]. This reduction of the plastid membranes from four known for cryptophytes and from three in the original *Dinophysis* plastids suggests that the plastid of *Dinophysis* is a long-established and permanent acquisition from cryptophyte algae [137]. Taken together, these observations argue for a long-standing plastid replacement in *Dinophysis* and functioning of the phycobiliprotein antenna in the pigment apparatus of this genus.

## 8. Advantages and Disadvantages of the Phycobiliprotein Antennae and Loss of PBS

The pigment apparatus of various oxygenic photosynthetics is arranged uniformly, having two main parts: PS II and PS I. Pigment–protein complexes of PS II and PS I consist of RCs and their own, or core, antennae where Chl *a* is used. There are 36 (or 35) and 96 (or 95) molecules of core Chl *a* per RC in PS II and PS I, respectively. The high rate of sunlight absorption and charge separation in the RCs indicates that much more energy can be typically utilized per unit time. An additional antenna provides energy to each of the RCs from several tens and hundreds of new pigment molecules, removing the light limit of photosynthesis. Compared to the possible strategy of increasing the number of RCs per photosynthetic cell, the additional antenna saves the energy resources used to maintain photosynthesis. PS I and PS II complexes, with the exception of some minor differences, are conservative structures, whereas antennae are characterized by diversity. Antennae fall into two categories based on their molecular properties: water-soluble phycobiliproteins and membrane-bound Chl *a/b*- and Chl *a/c*-containing pigment–protein complexes [24,138,139].

### 8.1. Functional Advantages of PBS over Other Antennae

Light-harvesting antennae perform three main functions: (1) an increase in light gathering, primarily in spectral regions where the absorption capacities of Chl *a* are insufficient; (2) the transfer of absorbed energy to photoactive complexes of the two photosystems; and (3) the optimization of light gathering under possible changes in intensity and spectral conditions. The efficiency of interchromophore migration within an antenna and from the antenna as a whole to PS I and PS II complexes is particularly high, reaching 95–98% in all antenna types [140]. Different light-harvesting complexes, using the same mechanisms of energy transfers, have reached an optimum of functioning independently of each other [140,141].

The advantage of PBSs over other antennae is the better light absorption in the middle region of the visible spectrum, in the so-called “green window” of Chl *a* absorption. Chls *b* and *c* within Chl *a/b* and Chl *a/c* pigment–protein complexes have insignificant shortwavelength shifts relative to Chl *a* absorption and do not possess such properties. The presence of carotenoids in Chl *a/b*- and Chl *a/c*-proteins that absorb light at 450–500 nm only partially compensates for this deficiency. 

Due to various phycobilin chromophores and their multiple compositions in the phycobiliproteins with a diversification of light, capturing this type of antenna is incredibly diverse. The maximum spectral range of absorption is clearly in favor of the phycobilisome antennae. Indeed, cyanobacteria are adapted to a wider spectral light range than any other oxygenic photosynthetics. Red algae are better adapted than other algae to the spectral composition of light penetrating into the water throughout the photic zone of the ocean, from the shallows to the lower water layers, and due to the presence of R-Pes, they remain the only photosynthetics at the greatest depths at all [142]. Comparing PBSs having sets of several different phycobiliproteins with various phycobilins and Chl a/*b* or Chl a/*c*-containing antennae carrying only one different from the Chl *a* pigment, the membrane antennae in principle cannot have adequate properties [24]. Therefore, PBSs are the antennae optimal in terms of their spectral phylogenetic adaptation capabilities. The diversity of PBSs and their spectral benefits allow this antenna to adapt to any possible photosynthetic light niche.

### 8.2. Disadvantages of PBS Antennae

The reasons for replacing PBS by membrane antennae should be sought in the selective factors of energy expenditures for the maintenance of the photosynthetic apparatus. The energy loss in any antenna is inversely proportional to the number of migration acts, which is one of the reasons why the antenna cannot grow indefinitely. Excessive antenna sizes also have another danger—concentration quenching of excitation, since any defective pigment molecule can become the center of energy dissipation into heat with the interruption of delivery to RC. There is thus a contradiction between the requirements for maximum light gathering and the compactness of the antenna. 

In this respect, an important negative difference from Chl *a/b*- and Chl *a/c*-proteins is the low relative content of chromophore groups in phycobiliproteins. The most abundant of them (APC, PCs, and PEs) have one phycobilin chromophore per 160, 110, and 70 amino acid residues, respectively. In the membranous LHC proteins of higher plants, 14 Chl *a* and *b* molecules account for a total of 250 amino acid residues. Thus, the content of phycobilin chromophores per unit of polypeptide mass remains 4–9 times lower than that of Chl *a/b*- and Chl *a/c*-proteins due to covalent and noncovalent binding to the apoprotein, respectively [13]. PBS proteins account for up to 60% of the water-soluble protein or up to 20% of the total dry mass of cyanobacterial cells [12,141].

The noncovalent incorporation of Chl molecules into the amino acid microenvironments of the apoprotein creates more “pockets” for chromophore placement than the covalent binding of Cys residues and phycobilins. In addition, a separate group of lyases is required to covalently link phycobilin chromophores to the apoprotein, and as a result, several dozen genes are required for PBS self-assembly [142,143]. At the same time, the noncovalent bonding of Chl molecules to LHC polypeptides occurs autocatalytically, without the participation of additional enzymes [144,145]. These factors increase the energy consumption of PBS, while according to calculations [146,147], a photoautotrophic cell can spend no more than 1/3 of the energy received from the photosynthetic apparatus for its maintenance.

Why has such a covalent bond been preserved in the course of evolution? Unlike the planar molecules of chlorophylls and porphyrins, phycobilins are compounds with easily flexible geometry. The shape of the phycobilin molecule can change from being cyclic porphyrin-like through to pseudocyclic to being a fully elongated one. For free phycobilins in solution, the pseudocyclic form is energetically advantageous. In native apoproteins, a linear configuration of bilin chromophores is maintained due to the thioether bond, which is necessary to increase the extinction coefficient in the visible spectral region [145]. For example, due to geometrical changes, the extinction coefficient in the red absorption band of apoprotein-bound phycocyanobilin increases 2–3 times compared to free pigment in solution, namely, from 35 mM^−1^ cm^−1^ to 75–120 mM^−1^ cm^−1^, becoming larger than that of Chl *a* (64 mM^−1^ cm^−1^ in vivo). Other than covalent bonding, features of the PBS-based pigment apparatus may also have been unacceptable for the evolution of photosynthesis. With few exceptions [148], cyanobacterial and red algal cells contain a single type of PBS, which does not allow for separate regulation of the antennae belonging to PS I or PS II. In contrast, the LHC polypeptides belonging to PS I and PS II are different in their majority. Separate regulation of the antenna sizes creates conditions for a more economical balancing of light absorption by PS I and PS II. 

### 8.3. Role of Cell Wall Formation

An important factor in the disappearance of PBS was the formation of the plant cell wall. Oxygenic photosynthesis is inextricably linked to the photodissociation of water. Together with dissolved minerals, water became an unrestricted source of existence, as a consequence of which plant motility decreased and was gradually lost. The attachment to the substrate required the strengthening of the plant thallus and simultaneously increased the vulnerability in relation to the consumers. The resulting need for protection created a natural tendency for the emergence of the cell wall. Therefore, the formation of eukaryotic photosynthesis in numerous groups of algae proceeded in parallel with the development of the cell wall. The evolutionary landfall of plants completed this trend. The peptidoglycan layer surrounding the cyanobacterial cell is only a partial alternative. 

It is believed that approximately 1 billion years have passed since the appearance of the primary cell wall [149]. In various plant phyla, cell walls share a common evolutionary origin. Comparative biochemical analysis shows a constant complication of the wall in parallel with the appearance of numerous groups of algae [150]. The wall may be absent if unicellular algae retain motility, but it is an indispensable attribute of all green algae and higher plants [149]. Starting with mosses, the wall has become more complex, dividing into primary and secondary walls; the cellulose wall of flowering plants has the most complex molecular architecture, with up to 2500 genes controlling its creation and functioning [151], which is much more than the number of genes involved in the composition of the photosynthetic apparatus.

The qualitative consideration above shows that if significant energy expenditures of a photosynthetic microorganism are associated with cellulose biosynthesis, the preservation of the PBS is unprofitable. The second essential factor causing the transformation of the antenna is associated with the PS I/PS II ratio in chloroplasts. Eukaryotes require more CO_2_ assimilation than cyanobacteria to build the cell wall, and therefore, chloroplasts actively export synthesized carbohydrates to the cytoplasm. The slime polysaccharides, pectins, and the thin cell envelope which constitute the protection of cyanobacteria contain a much lower proportion of carbon than cellulose as the basis of the eukaryotic cell wall. The standard ratio of PS I to PS II in cyanobacteria and red algae is 3:1. In this case, due to the increased part of cyclic electron transport in PS I, active photophosphorylation can proceed without autotrophic carbon dioxide fixation. The ratio of two photosystems in plant chloroplasts is 1:1, indicating that the pigment apparatus underwent stoichiometric changes adequate to the increasing demand for CO_2_, as the changed ratio of PS I to PS II is due to the enlargement of linear electron transport between them. An increase in the PS II fraction in chloroplasts is reflected in the appearance of thylakoid granules, and the presence of the Chl *a/b*-protein promotes their formation and the compacted arrangement of photosynthetic membranes in the chloroplast [24,138,139].

Cyanobacteria and plastids of glaucophytic and red algae have no staked grana regions because PBS is a very bulky antenna (for comparison, hemidiscoidal PBSs are 2–3 times larger than ribosomes). Located on the outer surface of inner thylakoid membranes, PBSs fill the interthylakoid space. Scanning electron microscopy indicates that PBSs on the thylakoid surface are arranged practically without gaps and, under optimal conditions for photosynthesis, occupy the entire surface of the thylakoid membrane [110,129]. The formation of thylakoid granes is required to increase the PS II content and change the ratio with PS I to 1:1, and the corresponding increase in the number of PBSs bound to the PS II dimers becomes incompatible. 

## 9. Conclusions

The optimality principle in biology implies the consideration of evolutionary factors in their totality [152]. PBSs are preserved as antennae if their functional advantages prevail over the existing disadvantages. The reason for the replacement of PBSs in the course of evolution should be sought jointly in the optimization of the cell’s energy expenditure, in the complication of the thylakoid system of chloroplasts, and in the growing demand for carbon dioxide required for cellulose biosynthesis. Of the 30 × 10^9^ tons of carbon assimilated annually in the form of CO_2_ by higher plants, cellulose accounts for approximately one third [150]. While CO_2_ fixation enzymes constitute a significant part of the cell protein, the cellular costs of maintaining the photoautotrophic metabolism conflict with the costs of maintaining the pigment photosynthetic apparatus.

The conservation of phycobiliproteins in unicellular eukaryotes correlates with their motility and the underdevelopment of the wall. Indeed, glaucophytes are unicellular algae equipped with flagella. Cryptophytic algae also have flagella. The marine macrophyte red algae are attached to the substrate, but in many of them, the need to strengthen the thallus is achieved by calcification. Due to the weak development of cellulose, the size of red algae is limited to a few tens of centimeters, while thalloms of, for example, brown algae can reach tens of meters [142].

The replacement of PBS by more compact chlorophyll *a/b*- and chlorophyll *a/c*-proteins was the evolutionary path that photosynthetic organisms took in perfecting pigment compartments. Together with the excessive costs of phycobiliprotein biosynthesis, the above factors led to the elimination of PBSs in land plants and most groups of algae with all the perfections of PBS performing the direct antenna functions.

## Figures and Tables

**Figure 1 ijms-24-02290-f001:**
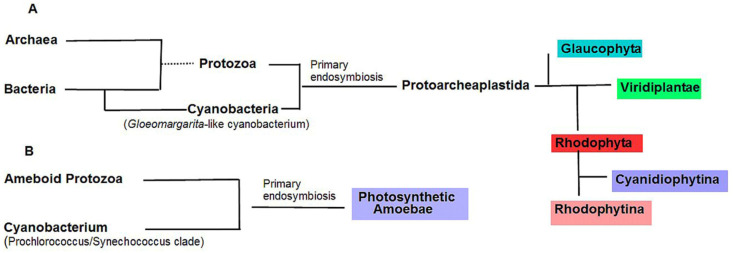
Scheme of primary endosymbioses in the origin of plastids. (**A**) The monophyletic origin of three phyla of Archaeplastida (Glaucophyta, Viridiplantae, and Rhodophyta). The length of the segments connecting the groups in sequence (from left to right) corresponds to the timing of their appearance. The Rhodophyta branch consists of the early isolated subphyla of microalgae Cyanidiophytina and subphyla Rhodophytina [7] containing some micro- and all macrophyte algae. (**B**) A later independent endosymbiosis that led to the appearance of photosynthetic amoebae.

**Figure 2 ijms-24-02290-f002:**
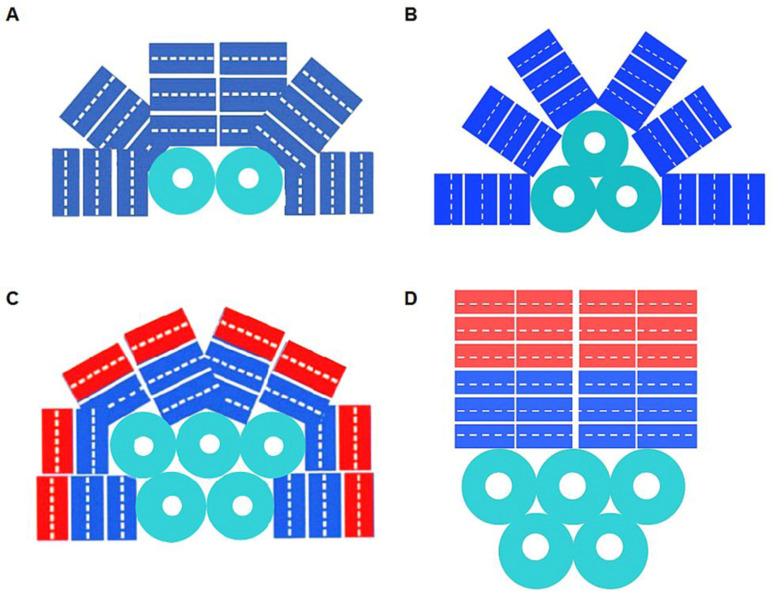
Morphological types of cyanobacterial PBSs. (**A**) Hemidiscoidal PBS with the bicylindrical core. (**B**) The most typical hemidiscoidal PBS with the tricylindrical core and six lateral cylinders. (**C**) PBS with the pentacylindrical core and eight lateral cylinders. (**D**) Bundle-shaped PBS (*Gloeobacter* type). APC —light blue; PC —blue; PE —red color.

**Figure 3 ijms-24-02290-f003:**
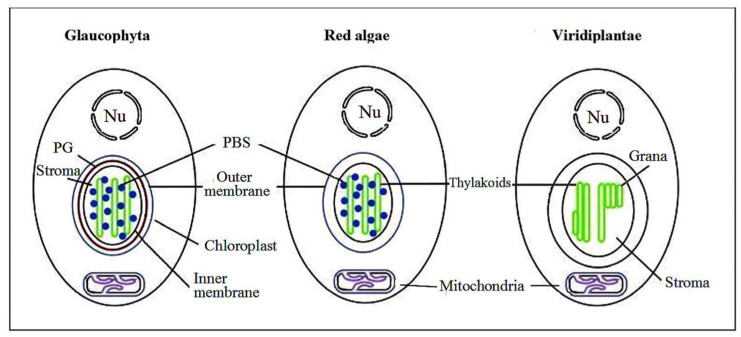
Three types of Archaeplastida double membrane chloroplasts. In glaucophytic algae (as well as in photosynthetic amaoebae), the chloroplast retains PBSs and the peptidoglycan layer (PG). Red algae retain PBSs, but the peptidoglycan layer is eliminated. In plastids of Viridiplantae, PBSs and peptidoglycan layers are lost while thylakoids form grana regions (similarly to Prochlorophyta).

**Figure 4 ijms-24-02290-f004:**
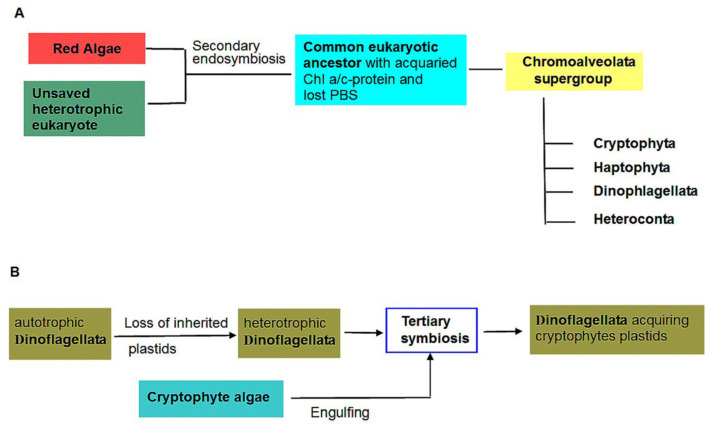
Schemes of secondary and tertiary endosymbioses. (**A**) Red line of secondary endosymbiosis, leading to a branched group of Chromoalveolata, including cryptophyte algae. (**B**) Tertiary endosymbiosis with the demonstration of replacement Dinoflagellata chromoplasts by plastids of engulfing cryptophyte alga.

**Table 1 ijms-24-02290-t001:** Phycobiliproteins in various photosynthetic clades (except Cryptophyta).

Phycobiliprotein	Presence in Photosynthetics
Allophycocyanin (PC, or APC)	Cyanobacteria, red, glaucophytic algae, and photosynthetic amoebae
C-phycocyanin (PC, or C-PC)	Cyanobacteria, glaucophytic algae, photosynthetic amoebae, and some red algae
Phycoerythrocyanin (PEC)	Some cyanobacteria
R-phycocyanin (R-PC, or R-PC I)	Red algae
R-PC II, R-PC III, R-PC IV and R-PC V	Cyanobacteria
C-phycoerythrin (C-PE)	Cyanobacteria
CU-phycoerythrin (CU-PE)	Cyanobateria
B- and b-phycoerythrins (B-PE and b-PE)	Red algae
R-phycoerythrin (R-PE)	Red algae

Abbreviations of phycobiliprotein names are given in brackets: C-PE is the common name for two cyanobacterial phycoerythrins having 5 or 6 PEB chromohores on (*αβ*)_1_ monomers; CU-PE is the common name for several cyanobacterial phycoerythrins having PUB and PEB chromophores in various ratios; B-PE and b-PE are the names of a variety of red algal PEs with (B-PE) or without (b-PE) *γ*-subunit in (*αβ*)_6_*γ* hexamers; R-PE is the name of the main red algal PE obligatory having *γ* subunit and PUB and PEB chromophores. PE I and PE II are other types of abbreviations for PEs with five and six phycobilin groups on (*αβ*)_1_ monomers, correspondingly. The table is based on the data collected in [11,12], with some additions and clarifications.

## Data Availability

Not applicable.

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
