# Peer review of "Phycobilisomes and Phycobiliproteins in the Pigment Apparatus of Oxygenic Photosynthetics: From Cyanobacteria to Tertiary Endosymbiosis"

_ijms, 2023, doi:10.3390/ijms24032290_

Round 1
Reviewer 1 Report
1. Line 35. What is Gloeomargarita []?
2. In the introduction, the novelty of this draft should be emphasized.
3. Attention should be paid to the layout of draft, such as line 221 and 252-255.
Author Response
Answer to Reviewer 1.
Dear Reviewer 1!
Thank you for your trouble in reading the manuscript and for the comments you made to improve it. We have tried to take them into account and make appropriate changes in the draft.
Here are our responses:
- Line 35. What is Gloeomargarita [ ]?
Superfluous unnecessary parenthesis deleted.
`In the Introduction, the novelty of this draft should be emphasized.
In the Introduction we have now outlined the goals of our review as follows:
“The purpose of this review was to summarize and consider all known cases of phycobiliprotein antennae in relation to the evolutionary origin of different groups of oxygenic photosynthetics. We also outline possible reasons why phycobiliproteins are absent in terrestrial plants and many groups of algae.”
- Attention should be paid to the layout of draft, such as line 221 and 252-255.
The subtitle on line 221 is rewritten as: Chlorophyll f containing cyanobacteria and their involvement in the process of photoacclimation to far red light
The original phrase on lines 252-255 was:
To summarize somewhat, we can state that changes in the composition of phycobilproteins occur differently in cyanobacteria possessing Chls b, d, or f biosynthesis. There is complete loss of PBSs and phycobiliproteins, a reduction from whole PBS to phycobiliprotein cylinders, or only a partial rearrangement of the PBS core and replacement of individual phycobiliproteins in the lateral cylinders.
It is rewritten as:
To summarize somewhat, it is stated that changes in the composition of phycobiliproteins occur differently in cyanobacteria possessing various kinds of Chl pigments. There is complete loss of PBSs and phycobiliproteins in Chl b-containg phototrophs, a reduction from whole PBS to phycobiliprotein cylinders in Chl d-containing species and only a partial rearrangement of the PBS core and replacement of individual phycobiliproteins in Chl f-containing cyanobacterial cells.
Besides, we have improved the text for English language according to comments of Ref’s 2 and 3.

Reviewer 2 Report
The manuscript is scientifically sound. It has potentially high interest to readers interested in the area of antimicrobial resistance. I would suggest that the authors consider the following point as they revise their manuscript.
1. English changes required.
Rest is good,
Author Response
Answer to Reviewer 2
Dear Reviewer 2!
Thank you for your trouble in reading the manuscript and for positive evaluation of our work.
We have improved the text for English language according to your comment and analogous recommendations of Ref. 3.
Reviewer 3 Report
In this manuscript the authors conducted a major literature review to consider the features of PBS-68 antenna in the listed groups of photosynthetics. I consider that it is a very good document, it is very well written and presented, it presents valuable information and it has a common thread that allows us to understand the content of the document very well. It is a very valuable manuscript with great contributions to knowledge. I made small contributions directly on the pdf that can help improve the content, I mainly suggest including several references and more updated ones, also including a chapter of conclusions. The manuscript is very good and can be published.

Author Response
Answers to Reviewer 3
Dear Reviewer 3!
Thank you for your trouble in reading the manuscript and for positive evaluation of our work. We have tried to take into account your comments to improve the text and have made appropriate changes in the draft.
Line 35. Reference or delete.
Extra parenthesis is deleted.
Figure 1. the background in light blue.
The ground is made lighter.
Line 61. references
References are added.
Line 62. Match this paragraph with the previous one.
The paragraph is matched.
Line 65. references
Reference is added.
Line 69. The problem or gap in knowledge that this review aims to solve should be clearer
We have added two clarifying the problem phrases:
The purpose of this review was to summarize and consider all known cases of phycobiliprotein antennae in relation to the evolutionary origin of different groups of oxygenic photosynthetics. We also outline possible reasons why phycobiliproteins are absent in terrestrial plants and many groups of algae.
Line 72. Phycobiliproteins
The typo was corrected.
Line 77. references
Reference is added.
Line 214. Join with the previous paragraph
The paragraph is joined with the previous one.
Line 221. The text should be larger
Text of the subtitle is enlarged to:
Chlorophyll f containing cyanobacteria and their involvement in the process of photoacclimation to far red light
Line 228. ???
Erroneous word removed
Line 252. Write in third person
Written in third person
Line 616. references
Reference is added.
Line 620. Include a brief explanation
Brief explanation is included:
The slime polysaccharides, pectins, and the thin cell envelope which constitute protection of cyanobacteria contain a much lower proportion of carbon than cellulose as the basis of the eukaryotic cells wall.
Line 625. subscript
Subscription was done.
Line 628. Include references in this paragraph
Reference was added.
Line 639. I think it is important to include at least one paragraph with the main conclusions and perhaps the perspectives
Conclusion was added
Besides, we have improved the text for English language according to your comments.